# FedGCN: Convergence-Communication Tradeoffs in Federated Training of Graph Convolutional Networks

**Yuhang Yao**
Carnegie Mellon University
yuhangya@andrew.cmu.edu

**Weizhao Jin**
University of Southern California
weizhaoj@usc.edu

**Srivatsan Ravi**
University of Southern California
sravi@isi.edu

**Carlee Joe-Wong**
Carnegie Mellon University
cjoewong@andrew.cmu.edu

## Abstract

Methods for training models on graphs distributed across multiple clients have recently grown in popularity, due to the size of these graphs as well as regulations on keeping data where it is generated. However, the cross-client edges naturally exist among clients. Thus, distributed methods for training a model on a single graph incur either significant communication overhead between clients or a loss of available information to the training. We introduce the Federated Graph Convolutional Network (FedGCN) algorithm, which uses federated learning to train GCN models for semi-supervised node classification with fast convergence and little communication. Compared to prior methods that require extra communication among clients at each training round, FedGCN clients only communicate with the central server in one pre-training step, greatly reducing communication costs and allowing the use of homomorphic encryption to further enhance privacy. We theoretically analyze the tradeoff between FedGCN's convergence rate and communication cost under different data distributions. Experimental results show that our FedGCN algorithm achieves better model accuracy with 51.7% faster convergence on average and at least $100\times$ less communication compared to prior work[1].

## 1 Introduction

Graph convolutional networks (GCNs) have been widely used for applications ranging from fake news detection in social networks to anomaly detection in sensor networks (Benamira et al., 2019; Zhang et al., 2020). This data, however, can be too large to store on a single server, e.g., records of billions of users' website visits. Strict data protection regulations such as the General Data Protection Regulation (GDPR) in Europe and Payment Aggregators and Payment Gateways (PAPG) in India also require that private data only be stored in local clients. In non-graph settings, federated learning has recently shown promise for training models on data that is kept at multiple clients (Zhao et al., 2018; Yang et al., 2021). Some papers have proposed federated training of GCNs (He et al., 2021a; Zhang et al., 2021). Typically, these consider a framework in which each client has access to a subset of a large graph, and clients iteratively compute local updates to a semi-supervised model on their local subgraphs, which are occasionally aggregated at a central server. Figure 1(left) illustrates the federated node classification task of predicting unknown labels of local nodes in each client.

---

[1] Code in https://github.com/yh-yao/FedGCN

37th Conference on Neural Information Processing Systems (NeurIPS 2023).

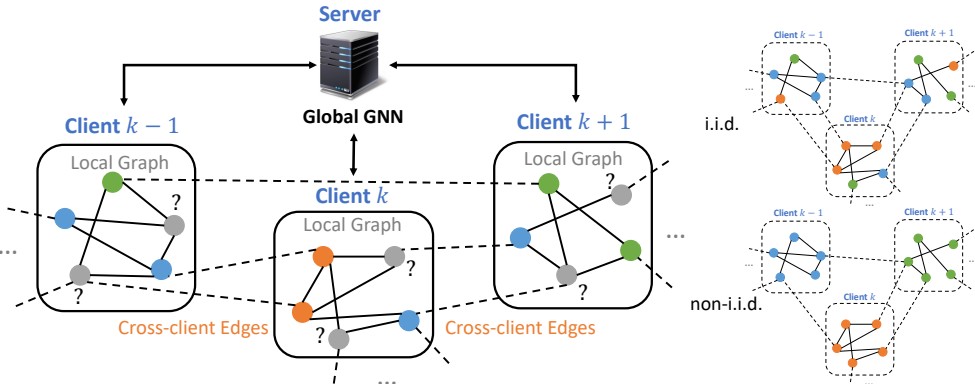

Figure 1: Federated GCN training schematic for node classification, with colors indicating the known labels of some nodes. Nodes in a graph (shown as circles) are distributed across clients, and dashed lines show cross-client edges between nodes at different clients. Arrows in the left figure indicate that each client can exchange updates with a central server during the training process to predict the unknown labels of the grey nodes in each client. At right, we show a graph with i.i.d (top) and non-i.i.d. (bottom) data distribution across clients, which affects the number of cross-client edges.

The main challenge of applying federated learning to GCN training tasks involving a single large graph is that cross-client edges exist among clients. In Figure 1, for example, we see that some edges will connect nodes in different clients. We refer to these as "cross-client edges".

Such cross-client edges typically are stored in both clients. Intuitively, this is due to the fact that edges are generated when nodes at clients interact with each other. Thus, the interaction record, though not personal node characteristics, is then naturally stored at both nodes, i.e., in both clients. For example, a graph may represent buying behaviors (edges) that exist between users (nodes) in two countries (clients). Users in one country want to buy items in another country. The records of these transactions between users in different countries (i.e., the cross-client edges) are then stored in both clients. Due to the General Data Protection Regulation, however, sensitive user information (node features including name, zip code, gender, birthday, credit card number, email address, etc.) cannot be stored in another country. Yet these cross-client edges cannot be ignored: including cross-country transactions (cross-client edges) is key for training models that detect international money laundering and fraud. Another example is online social applications like Facebook and LinkedIn. Users in different countries can build connections with each other (e.g., a person in the United States becoming Facebook friends with a person in China). The users in both the U.S. and China would then have a record of this friendship link, while the personal user information cannot be shared across countries.

However, GCNs require information about a node's neighbors to be aggregated in order to construct an embedding of each node that is used to accomplish tasks such as node classification and link prediction. Ignoring the information from neighbors located at another client, as in prior federated graph training algorithms (Wang et al., 2020a; He et al., 2021b), may then result in less accurate models due to loss of information from nodes at other clients.

Prior works on federated or distributed graph training reduce cross-client information loss by communicating information about nodes' neighbors at other clients in each training round (Scardapane et al., 2020; Wan et al., 2022; Zhang et al., 2021), which can introduce significant communication overhead and reveal private node information to other clients. We instead realize that the information needed to train a GCN only needs to be communicated once, before training. This insight allows us to further alleviate the privacy challenges of communicating node information between clients (Zhang et al., 2021). Specifically, we leverage Homomorphic Encryption (HE), which can preserve client privacy in federated learning but introduces significant overhead for each communication round; with only one communication round, this overhead is greatly reduced. Further, in practice each client may contain several node neighbors, e.g., clients might represent social network data in different countries, which cannot leave the country due to privacy regulations. Each client would then receive aggregated feature information about all of a node's neighbors in a different country, which itself can help preserve privacy through accumulation across multiple nodes. In the extreme case when nodes only have one cross-client neighbor, we can further integrate differential privacy techniques (Wei et al., 2020). We

propose the **FedGCN algorithm** for distributed GCN training based on these insights. FedGCN greatly reduces communication costs and speeds up convergence without information loss, compared with existing distributed settings (Scardapane et al., 2020; Wan et al., 2022; Zhang et al., 2021)

In some settings, we can further reduce FedGCN's required communication without compromising the trained model's accuracy. First, GCN models for node classification rely on the fact that nodes of the same class will have more edges connecting them, as shown in Figure 1(right). If nodes with each class tend to be concentrated at a single client, a version of the non-i.i.d. (non-independent and identically distributed) data often considered in federated learning, then ignoring cross-client edges discards little information, and FedGCN's communication round may be unnecessary. The model, however, may not converge, as federated learning may converge poorly when client data is non-i.i.d. (Zhao et al., 2018). Second, GCN models with multiple layers require accumulated information from nodes that are multiple hops away from each other, introducing greater communication overhead. However, such multi-hop information may not be needed in practice.

We analytically quantify the convergence rate of FedGCN with various degrees of communication, under both i.i.d. and non-i.i.d. client data. To the best of our knowledge, we are the first to analytically illustrate the resulting tradeoff between a fast convergence rate (which intuitively requires more information from cross-client edges) and low communication cost, which we do by considering a stochastic block model (Lei and Rinaldo, 2015; Keriven et al., 2020) of the graph topology. We thus quantify when FedGCN's communication significantly accelerates the GCN's convergence.

In summary, our work has the following **contributions:**

- We introduce FedGCN, an efficient framework for federated training of GCNs to solve node-level prediction tasks with limited communication and information loss, which also leverages Fully Homomorphic Encryption for enhanced privacy guarantees.

- We theoretically analyze the convergence rate and communication cost of FedGCN compared to prior methods, as well as its dependence on the data distribution. We can thus quantify the usefulness of communicating different amounts of cross-client information.

- Our experiments on both synthetic and real-world datasets demonstrate that FedGCN outperforms existing distributed GCN training methods in most cases with a fast convergence rate, higher accuracy, and orders-of-magnitude lower communication cost.

We outline related works in Section 2 before introducing the problem of node classification in graphs in Section 3. We then introduce FedGCN in Section 4 and analyze its performance theoretically (Section 5) and experimentally (Section 6) before concluding in Section 7.

## 2   Related Work

**Graph neural networks** aim to learn representations of graph-structured data (Bronstein et al., 2017). GCNs (Kipf and Welling, 2016), GraphSage (Hamilton et al., 2017), and GAT (Veličković et al., 2017) perform well on node classification and link prediction. Several works provide a theoretical analysis of GNNs based on the Stochastic Block Model (Zhou and Amini, 2019; Lei and Rinaldo, 2015; Keriven et al., 2020). We similarly adopt the SBM to quantify FedGCN's performance.

**Federated learning** was first proposed in McMahan et al. (2017)'s widely adopted FedAvg algorithm, which allows clients to train a model via coordination with a central server while keeping training data at local clients. However, FedAvg may not converge if data from different clients is non-i.i.d. (Zhao et al., 2018; Li et al., 2019a; Yang et al., 2021). We show similar results for federated graph training.

**Federated learning on graph neural networks** is a topic of recent interest (He et al., 2021a). Unlike learning tasks in which multiple graphs each constitute a separate data sample and are distributed across clients (e.g., graph classification (Zhang et al., 2018), image classification (Li et al., 2019b), and link prediction (Yao et al., 2023), FedGCN instead considers semi-supervised tasks on a single large graph (e.g., for node classification). Existing methods for such tasks generally ignore the resulting cross-client edges (He et al., 2021a). Scardapane et al. (2020)'s distributed GNN proposes a training algorithm communicating the neighbor features and intermediate outputs of GNN layers among clients with expensive communication costs. BDS-GCN (Wan et al., 2022) proposes to sample cross-client neighbors. These methods may violate client privacy by revealing per-node information to other clients. FedSage+ (Zhang et al., 2021) recovers missing neighbors for the input graph based

on the node embedding, which requires fine-tuning a linear model of neighbor generation and may not fully recover the cross-client information. It is further vulnerable to the data reconstruction attack, compromising privacy.

All of the above works further require communication at every training round, while FedGCN enables the private recovery of cross-client neighbor information with a single, pre-training communication round that utilizes HE. We also provide theoretical bounds on FedGCN's convergence.

## 3 Federated Semi-Supervised Node Classification

In this section, we formalize the problem of node classification on a single graph and introduce the federated setting in which we aim to solve this problem.

We consider a graph $\mathcal{G} = (\mathcal{V}, \mathcal{E})$, where $\mathcal{V} = [N]$ is the set of $N$ nodes and $\mathcal{E}$ is the set of edges. The graph is equivalent to a weighted adjacency matrix $\boldsymbol{A} \in \mathbb{R}^{N \times N}$, where $\boldsymbol{A}_{ij}$ indicates the weight of an edge from node $i$ to node $j$ (if the edge does not exist, the weight is zero). Every node $i \in \mathcal{V}$ has a feature vector $\boldsymbol{x}_i \in \mathbb{R}^d$, where $d$ represents the number of input features. Each node $i$ in a subset $\mathcal{V}^{train} \subset \mathcal{V}$ has a corresponding label $y_i$ used during training. Semi-supervised node classification aims to assign labels to nodes in the remaining set $\mathcal{V} \backslash \mathcal{V}^{train}$, based on their feature vectors and edges to other nodes. We train a GCN model to do so.

GCNs (Kipf and Welling, 2016) consist of multiple convolutional layers, each of which constructs a node embedding by aggregating the features of its neighboring nodes. Typically, the node embedding matrix $\boldsymbol{H}^{(l)}$ for each layer $l = 1, 2, \ldots, L$ is initialized to $\boldsymbol{H}^{(0)} = \boldsymbol{X}$, the matrix of features for each node (i.e., each row of $\boldsymbol{X}$ corresponds to the features for one node), and follows the propagation rule $\boldsymbol{H}^{(l+1)} = \phi(\boldsymbol{A}\boldsymbol{H}^{(l)}\boldsymbol{W}^{(l)})$. Here $\boldsymbol{W}^{(l)}$ are parameters to be learned, $\boldsymbol{A}$ is the weighted adjacency matrix, and $\phi$ is an activation function. Typically, $\phi$ is chosen as the softmax function in the last layer, so that the output can be interpreted as the probabilities of a node lying in each class, with ReLU activations in the preceding layers. The embedding of each node $i \in \mathcal{V}$ at layer $l + 1$ is then

$$\boldsymbol{h}_i^{(l+1)} = \phi \left( \sum_{j \in \mathcal{N}_i} \boldsymbol{A}_{ij} \boldsymbol{h}_j^{(l)} \boldsymbol{W}^{(l)} \right), \tag{1}$$

which can be computed from the previous layer's embedding $\boldsymbol{h}_j^{(l)}$ for each neighbor $j$ and the weight $\boldsymbol{A}_{ij}$ on edges from node $i$ to node $j$. For a GCN with $L$ layers in this form, the output for node $i$ will depend on neighbors up to $L$ steps away (i.e., there exists a path of no more than $L$ edges to node $i$). We denote this set by $\mathcal{N}_i^L$ (note that $i \in \mathcal{N}_i^L$) and refer to these nodes as $L$-hop neighbors of $i$.

To solve the node classification problem in **federated settings** (Figure 1), we consider, as usual in federated learning, a central server with $K$ clients. The graph $\mathcal{G} = (\mathcal{V}, \mathcal{E})$ is separated across the $K$ clients, each of which has a sub-graph $\mathcal{G}_k = (\mathcal{V}_k, \mathcal{E}_k)$. Here $\bigcup_{k=1}^K \mathcal{V}_k = \mathcal{V}$ and $\mathcal{V}_i \bigcap \mathcal{V}_j = \varnothing$ for $\forall i \neq j \in [K]$, i.e., the nodes are disjointly partitioned across clients. The features of nodes in the set $\mathcal{V}_k$ can then be represented as the matrix $\boldsymbol{X}_k$. The cross-client edges of client $k$, $\mathcal{E}_k^c$, for which the nodes connected by the edge are at different clients, are known to the client $k$. We use $\mathcal{V}_k^{train} \subset \mathcal{V}_k$ to denote the set of training nodes with associated labels $y_i$. The task of federated semi-supervised node classification is then to assign labels to nodes in the remaining set $\mathcal{V}_k \backslash \mathcal{V}_k^{train}$ for each client $k$.

As seen from (1), in order to find the embedding of the $i$-th node in the $l$-th layer, we need the previous layer's embedding $\boldsymbol{h}_j^{(l)}$ for all neighbors of node $i$. In the federated setting, however, some of these neighbors may be located at other clients, and thus their embeddings must be iteratively sent to the client that contains node $i$ for each layer at every training round. He et al. (2021a) ignore these neighbors, considering only $\mathcal{G}_k$ and $\mathcal{E}_k$ in training the model, while Scardapane et al. (2020); Wan et al. (2022); Zhang et al. (2021) require such communication, which may lead to high overhead and privacy costs. FedGCN provides a communication-efficient method to account for these neighbors.

## 4 Federated Graph Convolutional Network

In order to overcome the challenges outlined in Section 3, we propose our Federated Graph Convolutional Network (FedGCN) algorithm. In this section, we first introduce our federated training method with communication at the initial step and then outline the corresponding training algorithm.

**Federating Graph Convolutional Networks.** In the federated learning setting, let $c(i)$ denote the index of the client that contains node $i$ and $\boldsymbol{W}_{c(i)}^{(l)}$ denote the weight matrix of the $l$-th GCN layer of client $c(i)$. The embedding of node $i$ at layer $l+1$ is then $\boldsymbol{h}_i^{(l+1)} = \phi\left(\sum_{j \in \mathcal{N}_i} \boldsymbol{A}_{ij} \boldsymbol{h}_j^{(l)} \boldsymbol{W}_{c(i)}^{(l)}\right)$.

Note that the weights $\boldsymbol{W}_{c(i)}^{(l)}$ may differ from client to client, due to the independent local training in federated learning. For example, we can then write the computation of a 2-layer federated GCN as $\hat{\boldsymbol{y}}_i = \phi\left(\sum_{j \in \mathcal{N}_i} \boldsymbol{A}_{ij} \phi\left(\sum_{m \in \mathcal{N}_j} \boldsymbol{A}_{jm} \boldsymbol{x}_m^T \boldsymbol{W}_{c(i)}^{(1)}\right) \boldsymbol{W}_{c(i)}^{(2)}\right)$. To evaluate this 2-layer model, it then suffices for the client $k = c(i)$ to receive the message $\sum_{m \in \mathcal{N}_j} \boldsymbol{A}_{jm} \boldsymbol{x}_m^T$. We can write these messages as

$$\sum_{j \in \mathcal{N}_i} \boldsymbol{A}_{ij} \boldsymbol{x}_j, \text{ and } \left\{\sum_{m \in \mathcal{N}_j} \boldsymbol{A}_{jm} \boldsymbol{x}_j\right\}_{j \in \mathcal{N}_i / i}, \tag{2}$$

which are the feature aggregations of 1- and 2-hop neighbors of node $i$ respectively. This information does not change with the model training, as it simply depends on the (fixed) adjacency matrix $\boldsymbol{A}$ and node features $\boldsymbol{x}$. The client also naturally knows $\{\boldsymbol{A}_{ij}\}_{\forall j \in \mathcal{N}_i}$, which is included in $\mathcal{E}_k \bigcup \mathcal{E}_k^c$.

One way to obtain the above information is to receive the following message from clients $z$ that contain at least one two-hop neighbor of $k$:

$$\sum_{j \in \mathcal{N}_i} \mathbb{I}_z(c(j)) \boldsymbol{A}_{ij} \boldsymbol{x}_j, \text{ and } \forall j \in \mathcal{N}_i, \sum_{m \in \mathcal{N}_j} \mathbb{I}_z(c(m)) \cdot \boldsymbol{A}_{jm} \boldsymbol{x}_m. \tag{3}$$

Here the indicator $\mathbb{I}_z(c(m))$ is 1 if $z = c(m)$ and zero otherwise. More generally, for a $L$-layer GCN, each layer requires $\forall j \in \mathcal{N}_i^L / \mathcal{N}_i^{L-1}, \sum_{m \in \mathcal{N}_j} \mathbb{I}_z(c(m)) \cdot \boldsymbol{A}_{jm} \boldsymbol{x}_m$. Further, $\mathcal{E}_i^{L-1}$, i.e., the set of edges up to $L - 1$ hops away from node $i$, is needed for normalization of $A$.

To avoid the overhead of communicating between multiple pairs of clients, which can also induce privacy leakage when there is only one neighbor node in the client, we can instead send the aggregation of each client to the central server. In the 2-layer GCN example, the server then calculates the sum of neighbor features of node $i$ as $\sum_{j \in \mathcal{N}_i} \boldsymbol{A}_{ij} \boldsymbol{x}_j = \sum_{k=1}^K \sum_{j \in \mathcal{N}_i} \mathbb{I}_k(c(j)) \cdot \boldsymbol{A}_{ij} \boldsymbol{x}_j$. The server can then send the required feature aggregation in (2) back to each client $k$. Thus, we only need to send the accumulated features of each node's (possibly multi-hop) neighbors, in order to evaluate the GCN model. If there are multiple neighbors stored in other clients, this accumulation serves to protect their individual privacy[2]. For the computation of all nodes $\mathcal{V}_k$ stored in client $k$ with an $L$-layer GCN, the client needs to receive $\{\sum_{j \in \mathcal{N}_i} \boldsymbol{A}_{ij} \boldsymbol{x}_j\}_{i \in \mathcal{N}_{\mathcal{V}_k}^L}$, where $\mathcal{N}_{\mathcal{V}_k}^L$ is the set of $L$-hop neighbors of nodes $\mathcal{V}_k$.

FedGCN is based on the insight that GCNs require only the accumulated information of the $L$-hop neighbors of each node, which may be communicated in advance of the training. In practice, however, even this communication may be infeasible. If $L$ is too large, $L$-hop neighbors may actually consist of the entire graph (social network graphs have diameters $< 10$), which might introduce prohibitive storage and communication requirements. Thus, we design FedGCN to accommodate three types of communication approximations, according to the most appropriate choice for a given application:

- **No communication (0-hop):** If any communication is unacceptable, e.g., due to overhead, each client simply trains on $\mathcal{G}_k$ and ignores cross-client edges, as in prior work.
- **One-hop communication:** If some communication is permissible, we may use the accumulation of feature information from nodes' 1-hop neighbors, $\{\sum_{j \in \mathcal{N}_i} \boldsymbol{A}_{ij} \boldsymbol{x}_j\}_{i \in \mathcal{V}_k}$, to approximate the GCN computation. 1-hop neighbors are unlikely to introduce significant memory or communication overhead as long as the graph is sparse, e.g. social networks.
- **Two-hop communication:** To further improve model performance, we can communicate the information from 2-hop neighbors, $\{\sum_{j \in \mathcal{N}_i} \boldsymbol{A}_{ij} \boldsymbol{x}_j\}_{i \in \mathcal{N}_{\mathcal{V}_k}}$ and perform the aggregation of $L$-layer GCNs. As shown in Figure 2, the 2-hop approximation does not decrease model accuracy in practice compared to $L$-hop communication for $L$-hop GCNs, up to $L \leq 10$.

---

[2]In the extreme case when the node only has one neighbor stored in other clients, we can drop the neighbor, which likely has minimal effect on model performance, or add differential privacy to the communicated data.

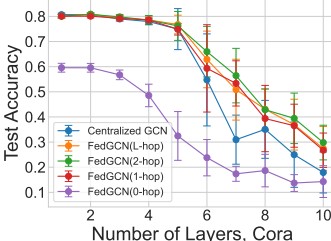 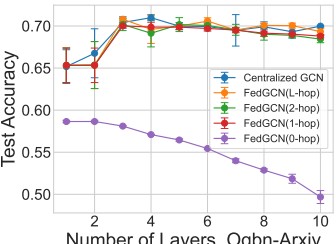

Figure 2: Test Accuracy of GCNs with different numbers of layers, using centralized and FedGCN training on Cora (left) and Ogbn-Arxiv (right) datasets with 10 clients and non-i.i.d partition. Communicating 2-hop information in FedGCN is sufficient for up to 10-layer GCNs. In Ogbn-Arxiv, BatchNorm1d is added between GCN layers to ensure consistent performance (Hu et al., 2020).

**Why Not $L$-hop Communication?** Although FedGCN supports $L$-hop communication, communicating $L$-hop neighbors across clients requires knowledge of the $L-1$ hop neighborhood graph structures, which may incur privacy leakage. If $L$ is large enough, $L$-hop neighbors may also cover the entire graph, incurring prohibitive communication and computation costs. Thus, in practice we restrict ourselves to 2-hop communication, which only requires knowledge of 1-hop neighbors. Indeed, Figure 2 shows that even on the Ogbn-Arxiv dataset, which has more than one million edges, adding more layers and $k$-hop communication does not increase model accuracy for $L \geq 3$ and $k \geq 2$. Thus, it is reasonable to use 0, 1, or 2-hop communication with $L \leq 3$-layer GCNs in practice.

**Secure Neighbor Feature Aggregation.** To guarantee privacy during the aggregation process of accumulated features, we leverage Homomorphic Encryption (HE) to construct a secure neighbor feature aggregation function. HE (Brakerski et al., 2014; Cheon et al., 2017) allows a computing party to perform computation over ciphertext without decrypting it, thus preserving the plaintext data.

The key steps of the process can be summarized as follows: (i) all clients agree on and initialize a HE keypair, (ii) each client encrypts the local neighbor feature array and sends it to the server, and (iii) upon receiving all encrypted neighbor feature arrays from clients, the server performs secure neighbor feature aggregation $\left[\!\left[\sum_{j \in \mathcal{N}_i} \boldsymbol{A}_{ij} \boldsymbol{x}_j\right]\!\right] = \sum_{k=1}^{K} \left[\!\left[\sum_{j \in \mathcal{N}_i} \mathbb{I}_k(c(j)) \cdot \boldsymbol{A}_{ij} \boldsymbol{x}_j\right]\!\right]$, where $[\![\cdot]\!]$ represents the encryption function. The server then distributes the aggregated neighbor feature array to each client, and (iv) upon receiving the aggregated neighbor feature array, each client decrypts it and moves on to the model training phase. We can also use HE for a secure model gradient aggregation function during the model's training rounds, which provides extra privacy guarantees.

Since model parameters are often floating point numbers and node features can be binary (e.g., one-hot indicators), a naïve HE scheme would use CKKS (Cheon et al., 2017) for parameters and integer schemes such as BGV (Brakerski et al., 2014) for features. To avoid the resulting separate cryptographic setups, we adopt CKKS with a rounding procedure for integers and also propose an efficient HE file optimization, Boolean Packing, that packs arrays of boolean values into integers and optimizes the cryptographic communication overhead. The encrypted features then only require twice the communication cost of the raw data, compared to 20x overhead with general encryption.

**Training Algorithm.** Based on the insights in the previous section, we introduce the FedGCN training algorithm (details are provided in Appendix A):

- **Pretraining Communication Round** The algorithm requires communication between clients and the central server at the initial communication round.
  1. Each client $k$ sends its encrypted accumulations of local node features, $\left[\!\left[\{\sum_{j \in \mathcal{N}_i} \mathbb{I}_k(c(j)) \cdot \boldsymbol{A}_{ij} \boldsymbol{x}_j\}_{i \in \mathcal{V}_k}\right]\!\right]$, to the server.
  2. The server then accumulates the neighbor features for each node $i$, $\left[\!\left[\sum_{j \in \mathcal{N}_i} \boldsymbol{A}_{ij} \boldsymbol{x}_j\right]\!\right] = \sum_{k=1}^{K} \left[\!\left[\sum_{j \in \mathcal{N}_i} \mathbb{I}_k(c(j)) \cdot \boldsymbol{A}_{ij} \boldsymbol{x}_j\right]\!\right]$.
  3. Each client receives and decrypts the feature aggregation of its one-hop, $\left[\!\left[\{\sum_{j \in \mathcal{N}_i} \boldsymbol{A}_{ij} \boldsymbol{x}_j\}_{i \in \mathcal{V}_k}\right]\!\right]$, and if needed two-hop, neighbors $\left[\!\left[\{\sum_{j \in \mathcal{N}_i} \boldsymbol{A}_{ij} \boldsymbol{x}_j\}_{i \in \mathcal{N}_{\mathcal{V}_k}}\right]\!\right]$.
- **Federated Aggregation** After communication, FedGCN uses the standard FedAvg algorithm McMahan et al. (2017) to train the models. Other federated learning methods, e.g., as proposed by Reddi et al. (2020); Fallah et al. (2020), can easily replace this procedure.

# 5 FedGCN Convergence and Communication Analysis

In this section, we theoretically analyze the convergence rate and communication cost of FedGCN for i.i.d. and non-i.i.d. data with 0-, 1-, and 2-hop communication.

We first give a formal definition of the i.i.d. and non-i.i.d. data distributions, using distribution-based label imbalance (Hsu et al., 2019; Li et al., 2022). Figure 3 visualizes eight example data distributions across clients. For simplicity, we assume the number of clients $K$ exceeds the number of node label classes $C$, though Section 6's experiments support any number of clients. We also assume that each class contains the same number of nodes and that each client has the same number of nodes.

**Definition 5.1.** Each client $k$'s *label distribution* is defined as $[p_1 \quad p_2 \quad \ldots \quad p_C] \in \mathbb{R}^C$, where $p_c$ denotes the fraction of nodes of class $c$ at client $k$ and $\sum_c p_c = 1$.

**Definition 5.2.** Clients' data distributions are *i.i.d.* when nodes are uniformly and randomly assigned to clients, i.e., each client's label distribution is $[1/C \quad \ldots \quad 1/C]$. Otherwise, they are *non i.i.d.*

Non-i.i.d. distributions include that of McMahan et al. (2017) in which $p_i = 1 - p + \frac{p}{C}$ for some $i = 1, 2, \ldots, C$ and $\frac{p}{C}$ otherwise, where $p \in [0, 1]$ is a control parameter; or the Dirichlet distribution (Hsu et al., 2019) $Dir(\beta/C)$, where $\beta \geq 0$ is a control parameter. With these distributions, each client has one dominant class. If $p = 0$ or $\beta \to \infty$, all nodes at a client have the same class.

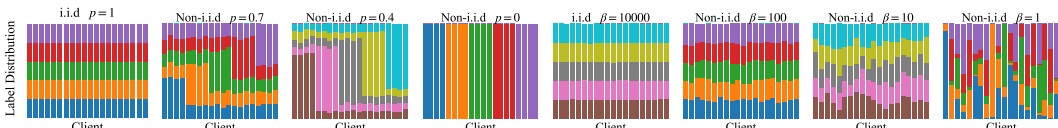

Figure 3: Example client data distributions. Different colors represent different node label classes. Client data is generated from the data distribution with different parameters $p$ and $\beta$ respectively.

We use $\|\boldsymbol{x}\|$ to denote the $\ell_2$ norm if $\boldsymbol{x}$ is a vector, and the Frobenius norm if $\boldsymbol{x}$ is a matrix. Given model parameters $\boldsymbol{w}$ and $K$ clients, we define the local loss function $f_k(\boldsymbol{w})$ for each client $k$, and the global loss function $f(\boldsymbol{w}) = \sum_{k=1}^{K} f_k(\boldsymbol{w})$, which has minimum value $f^*$.

**Assumption 5.3.** ($\lambda$-Lipschitz Continuous Gradient) There exists a constant $\lambda > 0$, such that $\|\nabla f_k(\boldsymbol{w}) - \nabla f_k(\boldsymbol{v})\| \leq \lambda \|\boldsymbol{w} - \boldsymbol{v}\|, \forall \boldsymbol{w}, \boldsymbol{v} \in \mathbb{R}^d$, and $k \in [K]$.

**Assumption 5.4.** (Bounded Global Variability) There exists a constant $\sigma_G \geq 0$, such that the global variability of the local gradients of the cost function $\|\nabla f_k(\boldsymbol{w_t}) - \nabla f(\boldsymbol{w_t})\| \leq \sigma_G, \forall k \in [K], \forall t$ [3]

**Assumption 5.5.** (Bounded Gradient) There exists a constant $G \geq 0$, such that the local gradients of the cost function $\|\nabla f_k(\boldsymbol{w_t})\| \leq G, \forall k \in [K], \forall t$.

Assumptions 5.3, 5.4 and 5.5 are standard in the federated learning literature (Li et al., 2019a; Yu et al., 2019; Yang et al., 2021). We consider a two-layer GCN, though our analysis can be extended to more layers. We work from Yu et al. (2019)'s convergence result for FedAvg[4] to find:

**Theorem 5.6.** *(Convergence Rate for FedGCN) Under the above assumptions, while training with $K$ clients, $T$ global training rounds, $E$ local updates per round, and a learning rate $\eta \leq \frac{1}{\lambda}$, we have*

$$\frac{1}{T} \sum_{t=1}^{T} \mathbb{E}\left[\|\nabla f(\boldsymbol{w_{t-1}})\|^2\right] \leq \frac{2}{\eta T}(f(\boldsymbol{w_0}) - f^*) + \frac{\lambda}{K}\eta \|I_{local} - I_{glob}\|^2 + 4\eta^2 E^2 G^2 \lambda^2, \quad (4)$$

*where $f^*$ is the minimum value of $f$, $I_{local} = K \boldsymbol{X}_k^T \boldsymbol{A}_k^T \boldsymbol{A}_k^T \boldsymbol{A}_k \boldsymbol{A}_k \boldsymbol{X}_k$, $I_{glob} = \boldsymbol{X}^T \boldsymbol{A}^T \boldsymbol{A}^T \boldsymbol{A} \boldsymbol{A} \boldsymbol{X}$.*

The convergence rate is thus bounded by the difference of the information provided by local and global graphs $\|I_{local} - I_{glob}\|$, which upper bounds the global variability $\|\nabla f_k(\boldsymbol{w}) - \nabla f(\boldsymbol{w})\|$. By 1- and 2-hop communication, the local graph $\boldsymbol{A}_k$ is closer to $\boldsymbol{A}$, resulting in faster convergence.

Table 1 bounds $\|I_{local} - I_{glob}\|$ for FedGCN trained on an SBM (stochastic block model) graph, in which we assume $N$ nodes with $C$ classes. Nodes in the same (different) class have an edge between them with probability $\alpha$ ($\mu\alpha$). Appendix D details the full SBM. Appendix H derives Table 1's

---

[3]Clients with i.i.d. label distributions may still have global variability $\sigma_G > 0$ due to having finite samples.
[4]Our analysis applies to any federated training algorithm with bounded global variability (Yang et al., 2021).

| | Non-i.i.d. | i.i.d. |
|---|---|---|
| 0-hop | $(1-\frac{1}{K^4})\frac{N^5}{C^5}\|B^4\| + (1-\frac{1}{C})^{\frac{5}{2}}(1-p)^5$ | $(1-\frac{1}{K^4})\frac{N^5}{C^5}\|B^4\|$ |
| 1-hop | $(1-\frac{1}{K^4}(1+c_\alpha p+c_\mu)^2)\frac{N^5}{C^5}\|B^4\| + (1-\frac{1}{C})^{\frac{5}{2}}(1-p)^5$ | $(1-\frac{1}{K^4}(1+c_\alpha+c_\mu)^2)\frac{N^5}{C^5}\|B^4\|$ |
| 2-hop | $(1-\frac{1}{K^4}(1+c_\alpha p+c_\mu)^6)\frac{N^5}{C^5}\|B^4\| + (1-\frac{1}{C})^{\frac{5}{2}}(1-p)^5$ | $(1-\frac{1}{K^4}(1+c_\alpha+c_\mu)^6)\frac{N^5}{C^5}\|B^4\|$ |

Table 1: Convergence rate bounds of FedGCN with the Stochastic Block Model and data distribution from McMahan et al. (2017) We define $c_\alpha = \frac{(1-\mu)\alpha N(K-1)}{CK}$ and $c_\mu = \frac{\mu\alpha N(K-1)}{K}$ for the SBM; $\alpha N$ is a constant, $c_\alpha \gg c_\mu$ and $c_\mu \simeq 0$. More hops speed up the convergence from the order of 2 to 6 (highlighted as purple and green). Communication helps more when data is more i.i.d with factor $c_\alpha p$ (highlighted as blue). Non-i.i.d. data implies a longer convergence time with factor $(1-p)^5$.

results; the SBM structure allows us to develop intuitions about FedGCN's convergence with different hops, simply by knowing the node features and graph adjacency matrix (i.e., without knowing the model). Appendix F derives corresponding expressions for the required communication costs. We validate these results in experiments and Appendix G, and make the following **key observations**:

- **Faster convergence with more communication hops**: 1-hop and 2-hop communication accelerate the convergence with factors $(1 + c_\alpha p + c_\mu)^2$ and $(1 + c_\alpha p + c_\mu)^6$, respectively. When the i.i.d control parameter $p$ increases, the difference among no (0-hop), 1-hop, and 2-hop communication increases: communication helps more when data is more i.i.d.

- **More hops are needed for cross-device FL**: When the number of clients $K$ increases, as in cross-device federated learning (FL), the convergence takes longer by a factor $\frac{1}{K^4}$, but 2-hop communication can recover more edges in other clients to speed up the training.

- **One-hop is sufficient for cross-silo FL**: If the number of clients $K$ is small, approximation methods via one-hop communication can balance the convergence rate and communication overhead. We experimentally validate this intuition in the next section.

# 6 Experimental Validation

We experimentally show that FedGCN converges to a more accurate model with less communication compared to previously proposed algorithms. We further validate Section 5's theoretical observations.

## 6.1 Experiment Settings

We use the Cora (2708 nodes, 5429 edges), Citeseer (3327 nodes, 4732 edges), Ogbn-ArXiv (169343 nodes, 1166243 edges), and Ogbn-Products (2449029 nodes, 61859140 edges) datasets to predict document labels and Amazon product categories (Wang et al., 2020b; Hu et al., 2020).

**Methods Compared. Centralized GCN** assumes a single client has access to the entire graph. **Distributed GCN** (Scardapane et al., 2020) trains GCN in distributed clients, which requires communicating node features and hidden states of each layer. **FedGCN (0-hop)** (Section 4) is equivalent to federated training without communication (**FedGraphnn**) (Wang et al., 2020a; Zheng et al., 2021; He et al., 2021a). **BDS-GCN** (Wan et al., 2022) randomly samples cross-client edges in each global training round, while **FedSage+** (Zhang et al., 2021) recovers missing neighbors by learning a linear predictor based on the node embedding, using cross-client information in each training round. It is thus an approximation of **FedGCN (1-hop)**, which communicates the 1-hop neighbors' information across clients. **FedGCN (2-hop)** communicates the two-hop neighbors' information across clients.

We consider the Dirichlet label distribution with parameter $\beta$, as shown in Figure 3. For Cora and Citeseer, we use a 2-layer GCN with Kipf and Welling (2016)'s hyper-parameters. For Ogbn-Arxiv and Ogbn-Products, we respectively use a 3-layer GCN and a 2-layer GraphSage with Hu et al. (2020)'s hyper-parameters. We average our results over 10 experiment runs. Detailed experimental setups and extended results, including an evaluation of the HE overhead, are in Appendix E and G.

## 6.2 Effect of Cross-Client Communication

We first evaluate our methods under i.i.d. ($\beta = 10000$) and non-i.i.d. ($\beta = 100, 1$) Dirichlet data distributions on the four datasets to illustrate FedGCN's performance relative to the centralized, BDS-GCN, and FedSage+ baselines under different levels of communication.

| | Cora, 10 clients | | | Citeseer, 10 clients | | |
|---|---|---|---|---|---|---|
| Centralized GCN | 0.8069±0.0065 | | | 0.6914±0.0051 | | |
| | $\beta = 1$ | $\beta = 100$ | $\beta = 10000$ | $\beta = 1$ | $\beta = 100$ | $\beta = 10000$ |
| FedGCN(0-hop) | 0.6502±0.0127 | 0.5958±0.0176 | 0.5992±0.0226 | 0.617±0.0118 | 0.5841±0.0168 | 0.5841±0.0138 |
| BDS-GCN | 0.7598±0.0143 | 0.7467±0.0117 | 0.7479±0.018 | 0.6709±0.0184 | 0.6596±0.0128 | 0.6582±0.01 |
| FedSage+ | 0.8026±0.0054 | 0.7942±0.0075 | 0.796±0.0075 | 0.6977±0.0097 | 0.6856±0.0121 | 0.688±0.0086 |
| FedGCN(1-hop) | **0.81±0.0066** | 0.8009±0.007 | 0.8009±0.0077 | **0.7006±0.0071** | 0.6891±0.0067 | 0.693±0.0069 |
| FedGCN(2-hop) | 0.8064±0.0043 | **0.8084±0.0051** | **0.8087±0.0061** | 0.6933±0.0067 | **0.6953±0.0069** | **0.6948±0.0032** |
| | Ogbn-Arxiv, 10 clients | | | Ogbn-Products, 5 clients | | |
| Centralized GCN | 0.7±0.0082 | | | 0.7058±0.0008 | | |
| | $\beta = 1$ | $\beta = 100$ | $\beta = 10000$ | $\beta = 1$ | $\beta = 100$ | $\beta = 10000$ |
| FedGCN(0-hop) | 0.5981±0.0094 | 0.5809±0.0017 | 0.5804±0.0015 | 0.6789±0.0031 | 0.658±0.0008 | 0.658±0.0008 |
| BDS-GCN | 0.6769±0.0086 | 0.6689±0.0024 | 0.6688±0.0015 | 0.6996±0.0019 | 0.6952±0.0012 | 0.6952±0.0009 |
| FedSage+ | 0.7053±0.0073 | 0.6921±0.0014 | 0.6918±0.0024 | 0.7044±0.0017 | 0.7047±0.0009 | 0.7051±0.0006 |
| FedGCN(1-hop) | 0.7101±0.0078 | **0.6989±0.0038** | 0.7004±0.0031 | 0.7049±0.0016 | **0.7057±0.0003** | **0.7057±0.0004** |
| FedGCN(2-hop) | **0.712±0.0075** | 0.6972±0.0075 | **0.7017±0.0081** | **0.7053±0.002** | **0.7057±0.0009** | 0.7055±0.0006 |

Table 2: Test Accuracy on four datasets, for i.i.d.($\beta = 10000$) and non-i.i.d. ($\beta = 100, 1$) data. FedGCN (1-hop,2-hop) performs best on i.i.d. and non-i.i.d. data, and FedGCN (0-hop) has the most information loss. We assume FedSage+'s linear approximator perfectly learns neighbor information.

As shown in Table 2, FedGCN(1-, 2-hop) has higher test accuracy than FedSage+ and BDS-GCN, reachiing the same test accuracy as centralized training in all settings. FedGCN(1-, 2-hop) is able to converge faster to reach the same accuracy with the optimal number of hops depending on the data distribution. In such cross-silo setting (10 clients), FedGCN(1-hop) achieves a good tradeoff between communication and model accuracy. FedGCN (0-hop) performs worst in the i.i.d. and non-i.i.d. settings, due to information loss from cross-client edges.

**Why Non-i.i.d Has Better Accuracy in 0-hop?** In Table 2 with 10 clients, 0-hop has better accuracy since non-i.i.d. has fewer cross-client edges (less information loss) than i.i.d as in theoretical analysis.

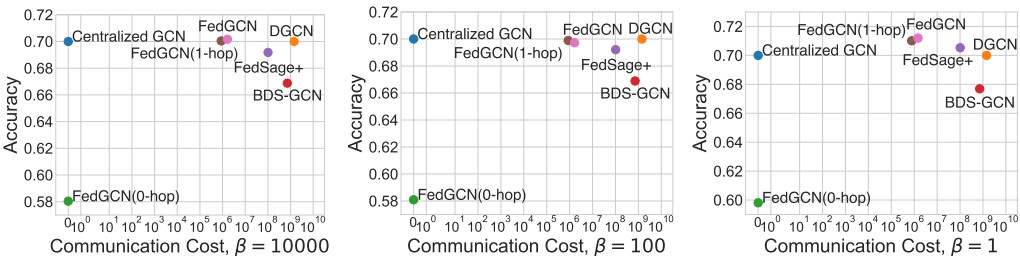

Figure 4: Test accuracy vs. communication cost until convergence of algorithms in the i.i.d., partial-i.i.d. and non-i.i.d. settings for the OGBN-ArXiv dataset. FedGCN uses orders of magnitude less communication (at least $100\times$) than BDS-GCN and FedSage+, while achieving higher test accuracy.

**Communication Cost vs. Accuracy.** Figure 4 shows the communication cost and test accuracy of different methods on the OGBN-ArXiv dataset. FedGCN (0-, 1-, and 2-hop) requires little communication with high accuracy, while Distributed GCN, BDS-GCN and FedSage+ require communication at every round, incurring over $100\times$ the communication cost of any of FedGCN's variants. FedGCN (0-hop) requires much less communication than 1- and 2-hop FedGCN, but has lower accuracy due to information loss, displaying a convergence-communication tradeoff. Both 1- and 2-hop FedGCN achieve similar accuracy as centralized GCN, indicating that a 1-hop approximation of cross-client edges is sufficient in practice to achieve an accurate model.

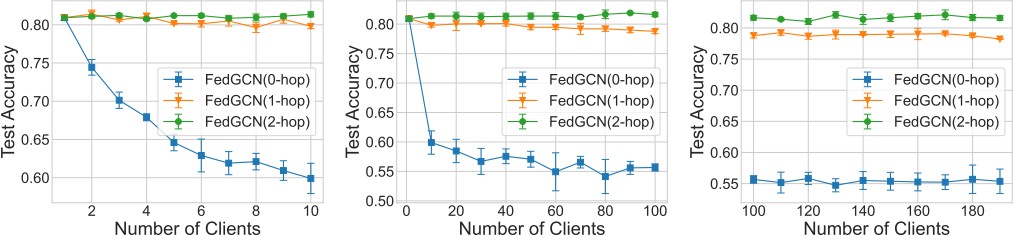

Figure 5: Test accuracy with the number of clients on the Cora dataset.

**Cross-Silo and Cross-Device Training.** As shown in Figure 5 (left), 1-hop and 2-hop communication achieve similar test accuracy in the cross-silo setting with few clients, though 1-hop communication has lower communication costs. However, in the cross-device setting with many clients (Figure 5 right), the 1-hop test accuracy drops with more clients, indicating that 2-hop communication may be necessary to maintain high model accuracy, as suggested by our theoretical analysis.

# 7 Conclusion & Future Directions

We propose FedGCN, a framework for federated training of graph convolutional networks for semi-supervised node classification. The FedGCN training algorithm is based on the insight that, although distributed GCN training typically ignores cross-client edges, these edges often contain information useful to the model, which can be sent in a single round of communication before training begins. FedGCN allows for different levels of communication to accommodate different privacy and overhead concerns, with more communication generally implying less information loss and faster convergence, and further integrates HE for privacy protection. We quantify FedGCN's convergence under different levels of communication and different degrees of non-i.i.d. data across clients and show that FedGCN achieves high accuracy on real datasets, with orders of magnitude less communication than previous algorithms.

Although the FedGCN can overcome the challenges mentioned above, it mainly works on training accumulation-based models like GCN and GraphSage. There are several open problems in federated graph learning that need to be explored.

**Federated Training of Attention-based GNNs** Attention-based GNNs like GAT (Graph attention network) require calculating the attention weights of edges during neighbor feature aggregation, where the attention weights are based on the node features on both sides of edges and attention parameters. The attention parameters are updated at every training iteration and cannot be simply fixed at the initial round. How to train attention-based GNNs in a federated way with high performance and privacy guarantees is an open challenge and promising direction.

**Neighbor Node and Feature Selection to Optimize System Performance** General federated graph learning optimizes the system by only sharing local models, without utilizing cross-device graph edge information, which leads to less accurate global models. On the other hand, communicating massive additional graph data among devices introduces communication overhead and potential privacy leakage. To save the communication cost without affecting the model performance, one can select key neighbors and neighbor features to reduce communication costs and remove redundant information. For privacy guarantee, if there is one neighbor node, it can be simply dropped to avoid private data communication. FedGCN can be extended by using selective communication in its pre-training communication round.

**Integration with $L$-hop Linear GNN Approximation methods** To speed up the local computation speed, $L$-hop Linear GNN Approximation methods use precomputation to reduce the training computations by running a simplified GCN ($A^L X W$ in SGC Wu et al. (2019), $[AXW, A^2XW, \ldots, A^L XW]$ in SIGN Frasca et al. (2020), and $\Pi X W$ in PPRGo Bojchevski et al. (2020) where $\Pi$ is the pre-computed personalized pagerank), but the communication cost is not reduced if we perform these methods alone. They are thus a complementary approach for efficient GNN training. FedGCN (2-hop, 1-hop) changes the model input ($A$ and $X$) to reduce communication in the FL setting, but the GCN model itself is not simplified. FedGCN can incorporate these methods to speed up the local computation, especially in constrained edge devices.

# 8 Acknowledgement

This research was supported by the National Science Foundation under grant CNS-1909306, Cloudbank support through CNS-1751075, and the Lee-Stanziale Ohana Fellowship by the ECE department at Carnegie Mellon University. The authors would like to thank Jiayu Chang, Cynthia (Xinyi) Fan, and Shoba Arunasalam for helping with the coding.

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
