# OpenReview forum: "FedGCN: Convergence-Communication Tradeoffs in Federated Training of Graph Convolutional Networks"
_NeurIPS.cc/2023/Conference — NeurIPS 2023 poster_

### Official Review · Reviewer_BmP4 · 2023-07-03

**Soundness:** 2 fair
**Presentation:** 2 fair
**Contribution:** 2 fair
**Rating:** 6
**Confidence:** 3

**Summary:**

The manuscript studies a semi-supervised node classification task using graph convolutional networks (GCNs). Specifically, motivated by the growth of the input graph (data) size, this paper considers a federated learning setting in which training is performed in a distributed manner, by partitioning the underlying graph and assigning each subgraph to a corresponding client. To deal with the communication overhead of the cross-client edges connecting graph nodes which need to be known for the target task, the paper proposes an algorithm that the central server first aggregates the information needed for each node in a client and then sends it to the designated node, rather than sending each information multiple times. The manuscript analyzes the tradeoff between the convergence rate of the algorithm and the communication overhead; in addition, a significant reduction on the communication cost with a reasonable accuracy is demonstrated through the experiment.

**Strengths:**

+ The problem is of sufficient interest since GNNs takes large-scale graphs as inputs, which can come with a heavy computational burden and may require huge storage costs.

**Weaknesses:**

+ It seems like the empirical results demonstrate that the proposed algorithm comes with a marginal gain compared to FedSage+ (Zhang et. al. 2021).

**Questions:**

+ The reviewer is wondering how the communication cost is measured in experimental validation (e.g., units).
+ Is the idea of aggregating cross-client edge information at the central server and sending the aggregated information (instead of sending each multiple times) to each client first proposed by this manuscript?

**Limitations:**

+ The assumption that the cross-client edges of client $k$, denoted by $\mathcal{E}_k^c$, are known to the client $k$ seems impractical due to the privacy issues.

---

> ### Author Rebuttal · Authors · 2023-08-09
>
> Thank you very much for your valuable comments!
>
> >**For weakness: marginal gain of accuracy compared to FedSage+**
>
> We agree that FedSage+ often achieves good accuracies. However, the main contribution of FedGCN is that it requires 100X less communication cost by using a pre-training communication round, without compromising accuracy. FedGCN can also ouperform FedSage+ in some settings. Ideally, FedSage+ can have a close performance to that of FedGCN(1-hop), as shown in Table 2. However, as in Fig. 5, FedGCN(2-hop) has 5% better accuracy than FedGCN(1-hop) (the upper bound of FedSage+) as the number of clients increases, which means FedGCN needs more hops of communication in such a cross-device setting to achieve high accuracy. We expect that it will outperform FedSage+ more dramatically in such a setting.
>
> >**Question: how the communication cost is measured?**
>
> It is calculated by the overall length of arrays needed to be sent, which is independent of the implementation. E.g. the client needs to send an array {1, 0, 1, 0} to the server, the communication cost is then 4. In this way, the cost can be independent of the implementation. Compression techniques may make the actual communication cost much lower than the array size. The code is also provided to reproduce.
>
> In Fig. 4. the communication cost of distributed GCN can be 10^9 for Ogbn-Arxiv, which works out to several GBs of communicated data. For Ogbn-Products (2,449,029 nodes), it is hundreds of GBs. FedGCN requires 100X less communication cost.
>
> In practical businesses (e.g. billions of nodes in Amazon and Facebook), the communication cost of distributed methods is prohibitive with extremely high latency at every training round. FedGCN's single pre-communication round can thus greatly improve performance.
>
> >**Question: Is the idea of server aggregation of cross-client edge first proposed by this manuscript?**
>
> Yes. A preliminary version of FedGCN required clients to communicate with each other to aggregate the cross-client edge information, which has privacy leakage. To resolve the issue, we designed the current server aggregation scheme after realizing that the server can perform such aggregation and the homomorphic encryption can also prevent privacy leakage on the server side.
>
> >**Cross-client edges are known to the client seems impractical**
>
> The cross-client edges typically exist in both clients, which we believe is common and practical. Intuitively, this is due to the fact that edges are generated when nodes at clients interact with each other. Thus, the interaction record, though not personal node characteristics, is then naturally stored at both nodes, i.e., in both clients. We provide some examples below and will use them to clarify this point in the paper’s introduction.
>
> For example, in Amazon, a graph may represent buying behaviors (edges) that exist between users (nodes) in two countries (clients). Users in one country want to buy items in another country. The records of these transactions between users in different countries (i.e., the cross-client edges) are then stored in both clients. Due to General Data Protection Regulation (GDPR), however, sensitive user information (node features including name, zip code, gender, birthday, credit card number, email address, etc.) cannot be stored in another country. Including cross-country transactions (cross-client edges) is key for training models that detect international money laundering and fraud.
>
> Another example is online social applications like Facebook and LinkedIn. Users in different countries can build connections with each other (e.g., a person in the United States becoming Facebook friends with a person in China). The users in both the U.S. and China would then have a record of this friendship link, while the personal user information cannot be shared across countries.

---

> > ### Comment · Reviewer_BmP4 · 2023-08-17
> >
> > I would like to thank the authors for addressing my questions and concerns.

---

> > > ### Author Response · Authors · 2023-08-18
> > >
> > > Dear Reviewer BmP4,
> > >
> > > Happy to know that we have addressed your questions and concerns!! We will add the discussion of the performance gain in cross-device settings and the above examples of cross-client edges to the paper!

---

### Official Review · Reviewer_z8YN · 2023-07-06

**Soundness:** 4 excellent
**Presentation:** 4 excellent
**Contribution:** 3 good
**Rating:** 7
**Confidence:** 3

**Summary:**

The authors propose a secure federated protocol over GCNs which are split across federated clients. To use the features of a node's neighbors present in other clients (of which both clients are mutually aware of an edge), the accumulation of neighbor features through the adjacency matrix is encrypted via some public key encryption scheme (such as homomorphic encryption), sent to the server, aggregated, and then sent back to the client, where it is decrypted and used to approximate the classical message passing GCN calculation. Feature accumulations of 0-hop, 1-hop, or 2-hop neighbor information may be used. The approach is studied over benchmarked over several well-known graph datasets and compared against other federated approaches, where the method (either 1-hop or 2-hop) is superior to non-FedGCN strategies.

**Strengths:**

1. Narrative is very nicely presented -- review of GCNs is nicely condensed and would be understandable for researchers outside of this particular area.

2. Algorithm is astoundingly simple, takes advantage of well-known public encryption schemes, protects client data features from one-another and from the server.

3. The method outperforms other federated algorithms over a wide array of benchmarks. There is apparently no need to accumulate anything beyond 2-hop features.

4. Convergence guarantee is provided with very standard non-convex, non-iid FL assumptions.

**Weaknesses:**

1. The one neighbor case wasn't adequately addressed. If the graph is heavily partitioned so there are many cross-client edges leading to a single neighbor, dropping the neighbor might not be appropriate. It was suggested differential privacy could be added to the single-neighbor accumulation but this will certifiably affect convergence, which would have been interesting to study.

2. The need for quantization/rounding via the encryption scheme was a bit rushed -- the decryption might not be perfect (see questions).

**Questions:**

1. If the encryption scheme requires rounding/quantization, this will affect the convergence rate as information transmission is imperfect. This is because the act of encryption-decryption is acting as a compression operator [1]. Many hallmark gradient compression algorithms study the effects of quantizing model parameters/updates on convergence, and it depends on the accuracy of the recovery. Is the encryption/decryption protocol simply assumed to be perfect? If not, can this be incorporated into the convergence guarantee?




[1] Stitch et al., 2018 "Sparsified SGD with Memory."

**Limitations:**

Discussed.

---

> ### Author Rebuttal · Authors · 2023-08-09
>
> We highly appreciate your opinions especially "Algorithm is astoundingly simple", which is also our goal in designing the algorithm.
>
> >**For weakness 1: the one neighbor case wasn't adequately addressed.**
>
> We agree that there could be cases when the graph is heavily partitioned and also nodes have one neighbor in other clients, although such cases are likely unusual in natural federated graphs. It is promising to potentially incorporate differential privacy in this extreme setup to provide an adequate privacy guarantee [1][2]. We are planning to include more discussions in the paper on tradeoffs between privacy and utilities from applying differential privacy.
>
> [1] Releasing Graph Neural Networks with Differential Privacy Guarantees. Transactions on Machine Learning Research 2023.
> [2] Federated learning with differential privacy: Algorithms and performance analysis. Transactions on Information Forensics and Security 2020.
>
> >**For weakness 2 and question 1: encryption scheme**
>
> As in Appendix G.2, the encrypted data can be exactly recovered and does not have a performance drop. The rounding we refer to here is merely a conversion from using floating point representation of binary values back to binary numbers. For example, we might have 1.000000000000003 in our CKKS scheme to represent the binary value of 1, where the error is introduced in the floating point representation and CKKS scheme errors, but the magnitude of these errors is usually negligible such that we can easily “round” them to their true values. This need of conversion is because we only use the CKKS scheme (approximation scheme for real numbers) in our system to avoid extra overheads and system complexity from using different HE schemes for different types of values (for example, using CKKS for real numbers while using BGV for integers or using FHEW for boolean values).

---

> > ### Comment · Reviewer_z8YN · 2023-08-14
> > **Further questions on encryption/decryption**
> >
> > Thanks to the authors for their responses. I have further follow-up regarding the encryption/decryption scheme:
> >
> > For the local model parameters (float64), under the CKKS scheme, even if we choose a scaling factor $\Delta$ which covers all the significant digits (let's say $\Delta=10^{16})$, the CKKS error still could distort the least significant bits such that the decryption is not perfect (but still extremely close, which concurs with your empirical observations), which should be accounted for in the analysis. You could assume the decryption is perfect (which is reasonable, for a large enough scaling factor).

---

> > > ### Author Response · Authors · 2023-08-14
> > > **Response to further questions on encryption/decryption**
> > >
> > > Dear Reviewer z8YN, thank you for the follow-up!!
> > >
> > > Your observation on the value error is correct and in general it has negligible impact on empirical evaluation as you also observed. However, we agree that adequate analysis on encryption/computation/decryption errors from CKKS scaling and other errors from other HE operators and phases can be included regarding the model performance, which will also be very helpful for the readers to understand the mechanism of HE. We plan to discuss it in the main paper (as you mentioned, the CKKS error still could distort the least significant bits such that the decryption is not perfect but still extremely close) and include such experiments in the appendix.

---

### Official Review · Reviewer_d6pK · 2023-07-10

**Soundness:** 3 good
**Presentation:** 3 good
**Contribution:** 3 good
**Rating:** 6
**Confidence:** 3

**Summary:**

The paper introduces a technique to facilitate the federated learning of graph convolution networks. The key idea is to send aggregated node features to each client before the federated learning. The approach involves transmitting these features to each client in an encrypted manner to ensure privacy.

Overall, I find the method proposed and the theoretical analysis interesting. The results look good. Some key details of the method might need to be clarified. I tend to accept this work.

**Strengths:**

- The paper proposes an improved framework for learning graph ConvNets federately. The results demonstrate the effectiveness.
- The theoretical convergence analysis is provided which illustrates a convegence-coomunite tradeoff.

**Weaknesses:**

Technical details
---
There are a few technical details that the authors may want to clarify for a better understanding of the proposed technique.

In equation (2), the authors claim that only feature aggregations of 1- and 2-hop neighbors of node $i$ are needed to evaluate the 2-layer GCN. It would be helpful if the authors could elaborate more on **why 1- and 2-hop aggregations are sufficient**. Is this an exact or approximate evaluation of the GCN model? Additionally, since GCN incorporates non-linearity in the model, it would be valuable to articulate how 2-hop feature aggregations can be directly used for evaluation.

In equation (3), regarding client $k$, it would be beneficial for the authors to clarify whether they compute 1-hop feature aggregations for all nodes $V$ or only for the nodes belonging to client $k$, denoted as $V_k$. In line 222, the authors mention $V_k$, but it seems that it should be $V$ instead. Otherwise, it appears impossible to obtain the cross-edge aggregation for node $i$.

Proof sketch
---
It might be beneficial to provide a sketch of the proof to demonstrate the idea of how to prove the bounds in Table 1. How does the number of hops involved in the analysis? Currently, results are up to 2-hops, is it possible to get results for the high numbers of hops?

**Questions:**

1. If the convergence-communication tradeoff is controlled by the number of hops, it appears that achieving a continuous or fine-grained tradeoff control might not be possible. What happens if the convergence achieved with 1-hop aggregations is poor while the communication required for 2-hop aggregations is excessive? How does the proposed technique address such scenarios?

2. Could you provide further details on how the communication costs are computed in Figure 4?

3. The author uses the term "cross-silo" to refer to a small number of clients and "cross-device" to refer to a large number of clients. It would be helpful to define the boundary between these two categories. How many clients are considered the minimum for classifying them as "cross-devices"?

**Limitations:**

Not discussed.

---

> ### Author Rebuttal · Authors · 2023-08-09
>
> Thank you very much for your recognition that the method and proofs are interesting! We are very happy to elaborate on more technical details and proof sketches.
> >**Technical details: why feature aggregations of only 1- and 2-hop neighbors of nodes are sufficient to evaluate the 2-layer GCN?**
>
> Based on Equation 1,
>
> $\mathbf{h}^{(l+1)}\_i=\phi\left(\sum_{j\in\mathcal{N}\_{i}}\mathbf{A}\_{ij}\mathbf{h}\_j^{(l)}\mathbf{W}^{(l)}\right), $
>
> as mentioned in line 127, for a GCN with L layers, the output for node i will depend on neighbors up to L steps away  (i.e., there exists a path of no more than L edges to node i). So feature aggregations of 2-hop neighbors give an exact evaluation for 2-layer GCN.
> >**Technical details: how 2-hop feature aggregations can be directly used for evaluation?**
>
> The 2-layer GCN computation for node $i$ is
> $$\mathbf{\hat{y}}\_i=\phi\left(\sum_{j\in\mathcal{N}\_i}\mathbf{A}\_{ij}\phi\left(\sum_{m\in\mathcal{N}\_j}\mathbf{A}\_{jm}\mathbf{x}_m^T \mathbf{W}^{(1)}\_{c(i)}\right)\mathbf{W}^{(2)}\_{c(i)}\right).$$
>
> $\sum_{m\in\mathcal{N}\_j}\mathbf{A}\_{jm}\mathbf{x}\_m^T$ includes both 1-hop feature aggregation $\sum_{j\in\mathcal{N}\_i} \mathbf{A}\_{ij}\mathbf{x}\_j$, and 2-hop feature aggregation
> $\left\\{\sum_{m\in\mathcal{N}\_j}\mathbf{A}\_{jm}\mathbf{x}\_j\right\\}\_{j\in\mathcal{N}\_{i}/i}$. The 2-hop feature aggregation is directly added in $\sum_{m\in\mathcal{N}\_j}\mathbf{A}\_{jm}\mathbf{x}\_m^T$. After completing the aggregation of this information, it then goes to the non-linear activation layer $\phi(\cdot)$.
> >**Technical details in equation 3: For client $k$, compute 1-hop feature aggregations for all nodes $V$ or only for the nodes $V_k$ belonging to client $k$?**
>
> As in line 222, each client $k$ sends its encrypted accumulations of local node features,
> $$\left[\left[ \left\\{\sum_{j\in\mathcal{N}\_i}\mathbb{I}\_k(c(j))\cdot\mathbf{A}\_{ij}\mathbf{x}\_j\right\\}\_{i\in{\mathcal{V}\_k}}\right]\right],$$
> to the server based on its local nodes $V_k$. The server then aggregates such information for all clients and the accumulated information for all nodes $V$. After that, the server selects the required 2-hop neighbor feature aggregation for each node $i$ and sends it back to the client $k$.
>
> >**Sketch of proof to demonstrate the idea of how to prove the bounds in Table 1**
>
> Step 1: As in Theorem 5.6, the analysis for general graphs relies on bounding the difference of the information provided by local and global graphs $\|I_{local}-I_{glob}\|$.
>
> Step 2: To further quantify such differences, we adopt the Stochastic Block Model (SBM) for analyzing the graph topology.
>
> Step 3: As shown in Appendix H.3.2, the information difference can be separated into two components: the difference in the number of nodes (different number of data samples) and the difference in node label distribution  (IID or Non-IID).
>
> Step 4: By calculating the two components under the SBM, we can then derive the equations (each one has two components) in Table 1.
> >**Proof Sketches: How does the number of hops involved in the analysis?**
>
> As in Theorem 5.6, Appendix H.1 (1-layer case) and Appendix H.2 (2-layer case), the difference of the information provided by local and global graphs $\|I_{local}-I_{glob}\|$ can be written as $\|K\mathbf{X}_k^T\mathbf{A}_k^T\mathbf{A}_k\mathbf{X}_k-\mathbf{X}^T\mathbf{A}^T\mathbf{A}\mathbf{X}\|$ for the 1-layer case and $\|K\mathbf{X}_k^T\mathbf{A}_k^T\mathbf{A}_k^T\mathbf{A}_k \mathbf{A}_k\mathbf{X}_k-\mathbf{X}^T\mathbf{A}^T\mathbf{A}^T\mathbf{A}\mathbf{A}\mathbf{X}\|$ for the 2-layer case.
>
> It can be extended to the L-hop case by expanding $\mathbf{A}^T\mathbf{A}$, though the computation (deriving the analytical form) will be increasingly tedious as the number of hops increases.
>
> We already have a cleaner version of the proof and we will also include the above proof sketch in the paper.
> >**Question 1: What happens if the convergence achieved with 1-hop aggregations is poor while the communication required for 2-hop aggregations is excessive?**
>
> This is an interesting future direction. The most natural way to make the communication tradeoff more granular is to perform neighbor sampling with probability $p$ to reduce the communication overhead [1]. Currently, we are communicating with all neighbors in the graph. We will include this direction as future work.
>
> [1] BNS-GCN: Efficient full-graph training of graph convolutional networks with partition-parallelism and random boundary node sampling.
> >**Question 2: Could you provide further details on how the communication costs are computed in Figure 4?**
>
> It is calculated by the overall length of arrays needed to be sent, which is independent of the implementation. E.g. the client needs to send an array {1, 0, 1, 0} to the server, the communication cost is then 4. In this way, the cost can be independent of the implementation. Compression techniques may make the actual communication cost lower than the array size, but their efficacy will depend on the specific data to be transmitted, and thus we do not consider them here. The code is also provided to reproduce.
>
> In Fig. 4. the communication cost of distributed GCN can be 10^9 for Ogbn-Arxiv, which works out to several GBs of communicated data. For Ogbn-Products (2,449,029 nodes), it is hundreds of GBs. FedGCN requires 100X less communication cost.
>
> In practical businesses (e.g. billions of nodes in Amazon and Facebook), the communication cost of distributed methods is prohibitive with extremely high latency at every training round. FedGCN's single pre-communication round can thus greatly improve performance.
> >**Question 3: How many clients are considered the minimum for classifying them as "cross-devices"?**
>
> We borrowed this terminology from [2], and there is no clear definition in the FL community. Generally speaking, "more than 100 clients" is cross-device, and "less than 100 clients" is cross-silo.
>
> [2] Federated Learning Tutorial, NeurIPS 2020

---

> > ### Comment · Reviewer_d6pK · 2023-08-14
> > **thank you for the rebuttal**
> >
> > Most of my questions are addressed by the rebuttal. I remain to be on the positive side for this submission.

---

> > > ### Author Response · Authors · 2023-08-14
> > >
> > > Dear Reviewer d6pK,
> > >
> > > Happy to know that our rebuttal has resolved your concerns!! We will add the discussed technical details and proof sketches to the paper!

---

### Official Review · Reviewer_PXRJ · 2023-07-27

**Soundness:** 2 fair
**Presentation:** 3 good
**Contribution:** 2 fair
**Rating:** 6
**Confidence:** 4

**Summary:**

The paper presents FedGCN, a framework designed for federated training of graph convolutional networks (GCNs) specifically for semi-supervised node classification. The proposed method aim to  communicate cross-client neighbor information just once before training initiates, diverging from previous methods that demanded communication in each round. This shift significantly reduces communication overhead and expedites convergence. FedGCN provides the flexibility to choose between 0-, 1-, or 2-hop neighbor communication to strike an optimal balance between overhead and model accuracy. Empirical results highlight FedGCN's effectiveness and minimal communication cost in comparison with previous techniques.

**Strengths:**

1. Reducing the training communication during distributed training of graph neural networks is a crucial problem when dealing with super large graphs.
2. The experiment is thorough, albeit with small-scale graph datasets.
3. The presentation is clear, well-structured, and easy to follow. I personally appreciate the results in Figure 3 and its presentation format.

**Weaknesses:**

1. The motivation is not clear. In the abstract, the motivation "keeping data where it is generated" and "a single connected graph cannot be disjointly partitioned onto multiple clients" is contradictory. We don't need to partition the graph; the graph itself has multiple partitions. If this is the case, where should we store the connection information for two nodes on different partitions since we need to "keep data where it is generated"? The idea of "keeping data where it is generated" also contradicts the proposed "X-hop communication". (Note that I accept the motivation that the graph is so large that it cannot be stored in one machine.)
2. The assumptions are not well grounded, well-explained, or empirically verified. The three assumptions - a) Lipschitz Continuous Gradient, b) Bounded Global Variability, and c) Bounded Gradient - are extremely strict. Even though this paper claims most of them are standard assumptions of FL, this is far from convincing to me. Since all the assumptions form the upper bound of the main claim (Equation (4)), this should be well addressed and treated.
3. The technical contribution may be limited. If I understand correctly, when rewriting the GCN equation to Equation (2), this paper removes the activation function in the first layer (otherwise, we would have to transform the hidden representation during training). This operation makes the proposal essentially a type of SGC [1]. If this is the case, the technical contribution may be limited.

4. The experiments appear to be quite weak.
- a. One of the motivations of this paper is the large size of the graph. However, the graphs used in the experiments are all small (even 'tiny' for Cora, CiteSeer).
- b. Since the graphs are small, the communication cost will be negligible.
- c. There are some important baselines missing. [2]


[1] Simplifying graph convolutional networks, ICML2019. \
[2] Learn Locally, Correct Globally: A Distributed Algorithm for Training Graph Neural Networks, ICLR 2022

**Questions:**

Please address my concerns in **Weaknesses**.

---

> ### Author Rebuttal · Authors · 2023-08-09
>
> Thank you for your helpful suggestions! We plan to incorporate them into our manuscript, as we detail below, and we believe that our reply will resolve your concerns.
>
> >**For weakness 1: motivation is not clear**
>
> “Keeping data where it is generated” refers to typical data constraints in federated settings where the graph data and its node features are naturally stored in the local clients, which preserves the privacy of users.
>
> In these settings, the cross-client edges typically exist in both clients. Intuitively, this is due to the fact that edges are generated when nodes at clients interact with each other. Thus, the interaction record, though not personal node characteristics, is naturally stored at both nodes, i.e., in both clients. We provide examples (Amazon and Facebook, see rebuttal to Reviewer BmP4) and will use them to clarify this point in the paper's introduction.
>
> The reviewer's comment that "keeping data where it is generated contradicts the proposed X-hop communication" exactly captures the main challenge of our paper: we believe that FedGCN's method of communication allows models to learn with cross-client edges, without revealing sensitive information across clients. "X-hop communication" is used to communicate averaged information with encryption rather than sending sensitive user features directly.
>
> In writing "a single connected graph cannot be disjointly partitioned onto multiple clients" we mean to say that the cross-client edges must exist. We agree that the sentence is a bit confusing. We will modify the abstract accordingly.
>
> As mentioned in Line 31, the main challenge of applying FL to GCN training involving a single large graph is that cross-client edges exist among clients.
> >**For weakness 2: the assumptions are not well-grounded, well-explained, or empirically verified**
>
> Lipschitz Continuous Gradient, Bounded Global Variability, and Bounded Gradient are standard assumptions for FL analysis [1-6]. For example, [6] incorporates them as assumption 1, assumption 2, and Eqn. 14 of Appendix B.2,
>
> $$\mathbb{E}[\mathcal{L}(\bar{\theta}^{t+1})]\leq\mathbb{E}[\mathcal{L}(\bar{\theta}^t)]+\mathbb{E}[\langle\nabla\mathcal{L}(\bar{\theta}^t),\bar{\theta}^{t+1}-\bar{\theta}^t\rangle]+\frac{L}{2}\mathbb{E}[\|\bar{\theta}^{t+1}-\bar{\theta}^t\|^2].$$
>
> They are “very standard non-convex, non-iid FL assumptions” (mentioned by reviewer z8YN (strength 4) and reviewer d6pK(strength 2)), and we believe they constitute the state-of-the-art FL convergence analysis.
>
> That said, we also agree that these non-convex assumptions are still relatively strict, since how to provide FL analysis without such assumptions is still an open problem. We will add discussion of these limitations to our paper. In brief, Bounded Global Variability and Bounded Gradient allow the convergence analysis to accommodate data distribution heterogeneity, a core challenge of FL [1-3]. The bounded gradient assumption in particular holds for certain activation functions, e.g., sigmoid functions, and bounded input features. Lipschitz Continuous Gradient is a technical condition on the shape of the loss function that is standard for non-convex analysis. It in fact relaxes the assumption of (strongly) convex loss functions that were previously common in analyzing FL and stochastic gradient descent [1]. Our theory is also based on [2]’s convergence result for FedAvg. We hope our theory can open the area of theoretical analysis of federated graph learning.
> >**For weakness 3: the technical contribution may be limited**
>
> We do not remove the activation function in the 1st layer. After getting the neighbor feature aggregation, the GCN computation is exactly the same as the centralized training given the same parameter $W$. Eqn. 2 shows only the aggregation of neighbor features; this aggregation is then passed into the activation function as usual in GCNs.
>
> As mentioned in our conclusion, the paper "Simplifying GCNs" aims to speed up the local computation by simplifying the GCN computation as $A^kXW$, which is an approximation of GCN. FedGCN can incorporate such methods to speed up local training.
> >**For weakness 4.a,4.b: graphs used in the experiments are all small, communication cost is then negligible**
>
> As mentioned in section 6.1, we experiment on Ogbn-ArXiv (169,343  nodes, 1,166,243 edges), and Ogbn-Products (2,449,029 nodes, 61,859,140 edges). We respectfully claim that a graph with 2,449,029 nodes and 61,859,140 edges is not “small”. [6] also uses Ogbn-ArXiv. We note that Ogbn-Products is bigger than all datasets used in [6].
>
> As described in Appendix E.3, experiments are done in a p3d.16xlarge instance with 8 GPUs (32GB memory for each GPU) and 10 g4dn.xlarge instances (16GB GPU memory in each instance). One run of the Ogbn-Products experiment can take 20 min by full-batch GPU graph training. CPU training is impossible in this case. Experiments take two weeks to finish all data points of Ogbn-Products. For communication cost, please refer to rebuttal to reviewer BmP4 in Question 2.
> >**For weakness 4.c: some important baselines missing**
>
> Thank you for mentioning the paper[6]. It proposes distributed graph training with global correction. For the global correction part, the server stores global graph information and node features, which causes serious privacy leakage in our federated setting. Although the paper studies a different setting, we will cite the paper and discuss it accordingly.
>
> [1] On the Convergence of FedAvg on Non-IID Data. ICLR 2019.
>
> [2] Parallel restarted SGD with faster convergence and less communication. AAAI 2019.
>
> [3] Achieving Linear Speedup with Partial Worker Participation in Non-IID FL. ICLR 2021.
>
> [4] Sharper convergence guarantees for asynchronous sgd for distributed and federated learning. NeurIPS 2022.
>
> [5] FeDXL: Provable FL for Deep X-Risk Optimization. ICML 2023.
>
> [6] Learn Locally, Correct Globally: A Distributed Algorithm for Training Graph Neural Networks, ICLR 2022

---

> > ### Comment · Reviewer_PXRJ · 2023-08-16
> > **Thanks for the rebuttal**
> >
> > I appreciate the author's response. However, I still have the following concerns:
> >
> > 1. Are these assumptions still valid for the graph? For instance, in graph federated learning, the $w$ and $v$ in Assumption 5.3 should be related since model $f(w)$ and $f(w)$ are trained with the connected nodes with high probability. How does this relationship impact the assumption? Additionally, how about Assumption 5.4 and Assumption 5.5? (I remain skeptical that such strict conditions can be assumed even on standard federated learning.)
> > 2. The datasets used in this paper are not real-world graphs that require "keeping data where it is generated." In other words, all the experiments in this paper are conducted on synthetic data. I find it hard to accept a paper that claims to address a real-world problem but only tests its solutions in simulated scenarios.

---

> > > ### Author Response · Authors · 2023-08-17
> > >
> > > Thank you for the new questions! We hope that our former rebuttal has addressed your prior concerns. We have done our best to answer your new questions below.
> > >
> > > >**Are these assumptions still valid for the graph?**
> > >
> > > For Assumption 5.3 (λ-Lipschitz Continuous Gradient),
> > >
> > >  $\|\nabla f_k(w)-\nabla f_k(v)\|\leq\lambda\|w-v\|$, $\forall w,v \in\mathbb{R}^d,$
> > >
> > > $w$ and $v$ in the statement of this assumption represent two arbitrary sets of parameter values of the model. In a graph neural network, $w$ represents the concatenation of parameters of each layer, i.e., $[W_1,W_2,...,W_L]$ where $W_l$ is the vectorized weight matrix of the $l$-th layer. For example, $w$ can be the model parameters at the first training iteration, and $v$ can be the model parameters after subsequent training iterations.
> > >
> > > The assumption is general for arbitrary $w$ and $v$. In words, it means that changing the model parameters from $w$ to $v$ will not change the value of the loss function $f_k$ by more than a constant factor, multiplied by the norm of $\left\|w-v\right\|$. The correlation between $w$ and $v$ will not affect the bound. Intuitively, one might in fact expect a correlation between $w$ and $v$ to make the Lipschitz property more likely to hold, since the loss value is less likely to change much if the new parameter values $(v)$ are correlated with the old parameter values $(w)$.
> > >
> > > Assumption 5.4, $\|\nabla f_k(w_t)-\nabla f(w_t)\|\leq\sigma_G$, follows from Assumption 5.5. If the local gradient is bounded, then since the global gradient is the sum of local gradients, it is also bounded. Thus, the difference between local and global gradients will also be bounded.
> > >
> > > For Assumption 5.5,
> > > $\|\nabla f_k(w_t)\|\leq G,$
> > > we agree that the bounded gradient assumption may not always hold. However, it can be shown that this assumption holds for certain activation functions, e.g., sigmoid functions, and bounded input features.
> > >
> > >
> > > >**Why do we adopt such assumptions?**
> > >
> > > Our analysis is based on that in [2]. To the best of our knowledge, all state-of-the-art FL papers, such as [1-6], make very similar assumptions. Since the main purpose of our paper is not to advance the convergence analysis of FL in general, but rather to show how this analysis applies to federated graph training, we follow these papers' assumptions. We believe that if better FL theory papers emerge that remove one or all assumptions, we can extend our work to this more advanced theory by analyzing the difference between the local gradient and global gradient in the graph setting.
> > >
> > > We further believe that, while our exact quantitative convergence bounds may not hold in practice given that some of the theoretical assumptions may be violated, the qualitative insights derived from those bounds may still be valuable. In Figure 5, for example, we empirically validate our qualitative observations on how FedGCN's convergence varies with the number of clients and number of hops. We will further emphasize in the paper that the convergence analysis suggests qualitative insights about FedGCN's performance, even if the exact mathematical expressions do not always hold.
> > >
> > > [1] On the Convergence of FedAvg on Non-IID Data. ICLR 2019.
> > >
> > > [2] Parallel restarted SGD with faster convergence and less communication. AAAI 2019.
> > >
> > > [3] Achieving Linear Speedup with Partial Worker Participation in Non-IID FL. ICLR 2021.
> > >
> > > [4] Sharper convergence guarantees for asynchronous sgd for distributed and federated learning. NeurIPS 2022.
> > >
> > > [5] FeDXL: Provable FL for Deep X-Risk Optimization. ICML 2023.
> > >
> > > [6] Learn Locally, Correct Globally: A Distributed Algorithm for Training Graph Neural Networks, ICLR 2022

---

> > > > ### Author Response · Authors · 2023-08-17
> > > >
> > > > >**Real-world graphs with "keeping data where it is generated."**
> > > >
> > > > We agree that ideally, we would evaluate FedGCN on real-world datasets for which it is important to keep data where it is generated for privacy reasons. However, we are not aware of any publicly available datasets with these characteristics. In fact, these very privacy concerns also present serious legal issues in making graph datasets publicly available, especially for big tech companies. Due to GDPR, for example, the data from the European Union cannot be sent to other countries outside EU boundary by default, let alone making it public. In order to make such datasets public and comply with these legal restrictions, the dataset must be highly anonymized, which is generally difficult, subject to several known privacy inference attacks, and may invite legal liability for a company if the anonymization process has errors or vulnerabilities.
> > > >
> > > > Due to the strict privacy requirements of many large graph datasets, experiment results on such datasets are also hard to publish, again due to legal concerns about leaking private information. We have in fact run experiments on such datasets, but we have not obtained permission to release these results. Due to the double blind submission, we do not give details on our experience here, though we are happy to answer follow-up questions within the double-blind constraints. To the best of our knowledge, no other papers on federated graph training (e.g., FedSage+, FedGraphNN, or BDS-GCN) utilize real-world graph datasets that contain privacy-sensitive data. All these federated graph training works use similar datasets (often the same datasets) as we do with manual data partition.
> > > >
> > > > A possible workaround to the privacy concerns is to create a natural partition, e.g., by country, of existing datasets like Ogbn-Arxiv and Ogbn-Products across clients, to better emulate conditions in which privacy concerns arise. However, Ogbn-Arxiv and Ogbn-Products do not have such geographical information, and we have been unable to find such a public graph dataset. One possible solution for research is creating an Arxiv citation graph dataset with institute information and partitioning the graph naturally based on the country. If necessary, we can create such a dataset and run experiments on it. In the present submission, we instead simulate many different data partition methods on the real-world Ogbn-Products datasets, which represent Amazon products, to make it closer to the real-world setting in which data is distributed across countries.
> > > >
> > > > We are also aware of some corporate efforts to open-source datasets that contain real-world graphs with natural partitions of data across clients, and we fully agree that experiments on these datasets would greatly strengthen federated graph training papers. However, to the best of our knowledge, such data are not yet available due to legal concerns. With our best effort here, we believe utilizing the public datasets are adequate to demonstrate the feasibility of our work under the privacy constraints mentioned above.

---

> > > > ### Comment · Reviewer_PXRJ · 2023-08-18
> > > > **Thank you for your rebuttal**
> > > >
> > > > I appreciate the detailed response to my concerns.
> > > >
> > > > 1. The response addressed my concerns about the dataset. I accept the reason that only using synthetic data.
> > > > 2. Regarding the response to my concerns about motivation: I don't think this is valid. If we store the edges of a node in one partition, we must also store the node from another partition. This seems contrary to your claim of “Keeping data where it is generated.”
> > > > 3. As you mentioned, "changing the model parameters from $w$ to $v$ will not change the value of the loss function $f$ by more than a constant factor, multiplied by the norm of $w-v$." I believe there might be an error here. Shouldn't it refer to the "gradient" instead of the "value of the loss function"?
> > > > 4. I don't question the validity of the Lipschitz property from a mathematical perspective since, if you choose a desired "constant factor", this will always hold. However, its real-world applicability seems highly unlikely. Thus the statement that the "Lipschitz property is more likely to hold" doesn't make sense to me.
> > > >
> > > > I am leaning toward the negative side of this submission at this time.

---

> > > > > ### Author Response · Authors · 2023-08-18
> > > > >
> > > > > Dear Reviewer PXRJ,
> > > > >
> > > > > Good to know that we have addressed your question about why we use real datasets with simulated data partitions. We appreciate your response and are happy to answer your further questions!
> > > > >
> > > > > >**If we store the edges of a node in one partition, we must also store the node from another partition.**
> > > > >
> > > > > Cross-client edges are naturally stored at both nodes, i.e., in both clients where the nodes reside. However, the personal node characteristics (node features) only exist in the local client. This is what we mean by “Keeping data where it is generated,” in the sense that the edge is generated at nodes in both clients, and therefore knowledge of the edge exists at both clients. However, since this phrase appears to have caused significant confusion, we will remove it from the paper and replace it with a more detailed explanation and example like the one below.
> > > > >
> > > > > One example is a graph encoding relationships on online social applications like Facebook and WhatsApp. Users in different countries can build connections with each other (e.g., a person in the United States becoming Facebook friends with a person in the European Union). The users in both the European Union and the U.S. would then have a record of this friendship link: for example, servers in the U.S. might have the user ID of the person in the E.U. with whom a U.S. person is a friend. However, personal user information (name, zip code, gender, birthday, credit card number, email address, etc.) cannot be shared across countries; thus, this information about the E.U. person would not be stored on U.S. servers. This node (user) information is used by the graph neural network for node classification.
> > > > >
> > > > > >**Shouldn't that sentence in rebuttal refers to the "gradient" instead of the "value of the loss function"**
> > > > >
> > > > > You are correct. It should be the gradient $\nabla f$ instead of $f$.
> > > > >
> > > > > >**Lipschitz property is not likely to hold in real-world applications**
> > > > >
> > > > > In this sentence, we mean, if the new parameter values are correlated with the old parameter values, the Lipschitz property is then more likely to hold.
> > > > >
> > > > > We fully agree that the Lipschitz property may not hold in real-world applications and that the Lipschitz upper bound can be loose, depending on the shape of the specific loss function and its gradient. However, to the best of our knowledge, state-of-the-art FL theoretical techniques do not address tighter bounds than the given Lipschitz condition. We just adopt such assumptions. As stated in our previous reply, we do believe that there are valuable qualitative insights that can emerge from this analysis. In Figure 5, for example, we empirically validate our qualitative observations on how FedGCN's convergence varies with the number of clients and number of hops.
> > > > >
> > > > > We believe that if better FL theory papers emerge that remove one or all assumptions, we can extend our work to this more advanced theory by analyzing the difference between the local gradient and global gradient in the graph setting.

---

> > > > > > ### Comment · Reviewer_PXRJ · 2023-08-20
> > > > > > **Thanks to your response**
> > > > > >
> > > > > > I appreciate the hard work of the authors in addressing my concerns.
> > > > > >
> > > > > > From my viewpoint, all the assumptions of FL in this paper are too strict for real-world applications (note that I do not want to invalidate the whole FL community; I just believe this kind of theoretical analysis is problematic and nonsensical). However, this particular issue should not be attributed solely to this paper. I highly recommend that the authors incorporate our discussions into the updated version and discuss the limitations further.
> > > > > >
> > > > > > I would like to raise my score from 3 to 6!
> > > > > >
> > > > > > Good luck.
> > > > > > Reviewer PXRJ

---

> > > > > > > ### Author Response · Authors · 2023-08-20
> > > > > > >
> > > > > > > Dear Reviewer PXRJ,
> > > > > > >
> > > > > > > Excited to know that we have addressed your concerns!! We highly appreciate your questions and suggestions, which will also be very helpful for the FL and graph community. We also agree that these common assumptions in the FL community are not practical enough. The basis is that the theoretical analysis of neural networks in ML still has a gap to practice. Many researchers are devoted to making theoretical analysis practical and we believe there will be better ML theory.
> > > > > > >
> > > > > > > We will add the discussion about cross-client edges and further discuss the limitations of these theoretical assumptions in the main paper.

---

### Decision · Program_Chairs · 2023-09-21

**Decision:**

Accept (poster)

**Comment:**

The paper is well written and made some interesting contribution for federated learning in graph neural networks. The authors are encouraged to polish the paper to clarify the motivation of the work, and clearly present and motivate their assumptions.